# Factors Affecting Progeny Production and Sex Ratio of *Gryon aetherium* (Hymenoptera: Scelionidae), a Candidate Biological Control Agent for *Bagrada hilaris* (Hemiptera: Pentatomidae)

**DOI:** 10.3390/insects13111010

**Published:** 2022-11-02

**Authors:** Evelyne Hougardy, Brian N. Hogg

**Affiliations:** Invasive Species and Pollinator Health Research Unit, USDA-ARS, Albany, CA 94710, USA

**Keywords:** biological control, egg parasitoids, oviposition behavior, rearing, sex ratio

## Abstract

**Simple Summary:**

Biological control programs using natural insect enemies often rely on the production of large numbers of insects for quarantine assessment and release. Many factors can influence the production of natural enemies, and the production of females is particularly important because females produce offspring. In a series of experiments, we studied the effects of several factors on the production of a parasitic wasp that attacks the eggs of an invasive stinkbug pest of cole crops (i.e., cabbage, broccoli, kale, etc.). The total production of offspring dropped when females were older than 24 days old and was highest when the temperature during oviposition was 26.6 °C. The production of females increased when the eggs were exposed to female wasps for one day rather than two days. The results show that the production of this parasitic wasp can be maximized when eggs are exposed for only one day, when females that are less than 24 days old are used, and at temperatures of around 26 °C.

**Abstract:**

Manipulating the factors that influence progeny production and sex ratio in parasitoids can help maximize the production of quarantine bioassays and/or mass releases. In a series of experiments, we studied the effects of several factors on offspring production and sex ratio in the parasitoid *Gryon aetherium* (Hymenoptera: Scelionidae), a candidate biological control agent for *Bagrada hilaris* (Hemiptera: Pentatomidae). Progeny production was influenced by maternal age and dropped when females were 24 or 28 days old and decreased on the second day of exposure. Overall, the offspring sex ratio was highly variable in *G. aetherium* and was affected by the duration of exposure, with higher proportions of females emerging after one day of exposure than after two days, but was not affected by female density, female age/host deprivation, or temperature during oviposition. Progeny production was affected by the temperature during oviposition and was highest at 26.6 °C. The results indicate that production of *G. aetherium* can be maximized at one day of exposure, using females that are less than 24 d old, and at temperatures of around 26 °C.

## 1. Introduction

Knowing what influences progeny production and sex ratio has practical implications in biological control programs using parasitoids because of the need to maximize female production for quarantine bioassays and/or mass production. Because of their haplodiploid genetic system and their ability to control the fertilization of their eggs during oviposition, hymenopteran parasitoid females can decide the sex of their offspring [1]: fertilized eggs (diploid) develop into females, while unfertilized (haploid) eggs develop into males (except for thelytokous parthenogenetic species where females develop from unfertilized eggs). Several factors have been identified to influence the maternal decision to fertilize an egg or not (reviewed by [2,3]). These factors can either be parental (e.g., maternal and/or paternal age) [4], host-related (e.g., host size and quality) [5], environmental (e.g., temperature) [6], and/or influenced by local mate competition (e.g., amount of mating at the emergence site and female density) [7]. While observed changes in the sex ratio in relation to one or several of these factors are adaptative and the result of females voluntarily adjusting the sex of their eggs, in some cases, a shift in the progeny sex ratio may also be the result of physiological constraints rather than behavioral changes [6,8].

Most of the factors influencing progeny production and sex ratio in parasitic wasps could be manipulated in laboratory settings to improve the production of females. For instance, maternal age is known to affect the progeny sex ratio with the proportion of female offspring usually decreasing with maternal age, probably as a consequence of sperm depletion or decreased sperm viability [9,10]. However, giving freshly emerged females a short period of host deprivation right after emergence can also have benefits, especially in synovigenic parasitoids, as it allows females to mature more eggs and increase their progeny production on the first day of oviposition [11]. Contrastingly, long periods with no access to hosts are usually detrimental to the overall fecundity and can lead to the production of more males [12]. Female density is another factor that has been extensively studied in the context of the local mate competition theory [13]. This theory assumes mating occurs at the emergence site and states that the presence of conspecific females on a host patch leads to an increase in the production of male progeny. When only one female parasitizes the host patch, she needs to produce just enough males to fertilize her female progeny. However, when the number of females exploiting the patch increases, it is advantageous for females to produce more sons that will mate with the daughters of other females. The duration of exposure to hosts could also affect the progeny sex ratio if it is linked to an increased risk of superparasitism (i.e., when females repeatedly oviposit in already-parasitized hosts) with time. In solitary parasitoids, when the resource provided by one host can only sustain the development of one individual, superparasitism inevitably leads to the mortality of supernumerary eggs/larvae, and, if that mortality is different based on the sex of the egg/larva [14], the result could be a shift in the sex ratio [2]. Temperature, generally extreme hot or cold, is another factor that can affect progeny production and the sex ratio through different mechanisms [2]. For instance, females could lay more sons at extreme temperatures because female fitness is more affected by high temperatures than male fitness [6]. Physiological constraints can also shift the progeny sex ratio because the extreme temperature may prevent egg fertilization during oviposition [6] or because of sex-specific differential survival at extreme temperatures [15,16].

The following study was motivated by the need to optimize production in a quarantine colony of *Gryon aetherium* Talamas (Hymenoptera: Scelionidae), a biological control candidate of *Bagrada hilaris* Burmeister (Hemiptera: Pentatomidae). The mean progeny sex ratio in *G. aetherium* quarantine colonies tends to be 45–46% females [17,18]. In general, *G. aetherium* appears to have a lower sex ratio compared to other *Gryon* spp. For instance, the overall progeny sex ratio in *G. gallardoi* (Brethes) is 79% females [19] and, in *G. pennsylvanicum* (Ashmead), is 90–93% females [20]. In addition, an adventive population of *G. aetherium* was recently discovered in California with an overall sex ratio of 59% females [17], suggesting that there is potential for sex ratio manipulation in the laboratory. After an assessment of the *G. aetherium* oviposition rate and sex allocation sequence, the effect of female density, the duration of exposure, the combined effect of maternal age and host deprivation, and the temperature during oviposition were investigated to better understand their effect on progeny production and sex ratio in our quarantine colony.

## 2. Materials and Methods

### 2.1. Insect Colonies

A colony of *B. hilaris*, initiated with insects collected in 2015–2018 in Yolo and Monterey counties in California, was held in a Bug Dorm cage (61 × 61 × 61 cm, BioQuip Products Inc, Rancho Dominguez, CA, USA) in the laboratory under 28–30 °C, 30–40% RH, and 14 L:10 D photoperiod. Nymphs and adults were fed organic broccoli florets (*Brassica oleracea* L.) and kale leaves (*Brassica oleracea* L.) supplemented by sweet alyssum (*Lobularia maritima* (L.) Desv.) and sunflower seeds (*Helianthus annuus* L.). Bagrada bug eggs used in rearing and experiments were produced in oviposition boxes (25 × 17 × 8 cm, ventilated plastic containers) containing two uncovered Petri dishes (50 mm diameter) filled with sieved sand (Quikrete Brown Play Sand, American Canyon, CA, USA) and sheltered under a cardboard tent (a square piece of white card, 80 × 80 mm, folded in half lengthwise). Oviposition boxes were regularly replenished with food and young adults from the main colony and eggs were harvested daily by sieving the sand using a No. 35 sieve (0.5 mm).

*G. aetherium*-parasitized *B. hilaris* eggs were collected in Pakistan in 2016 and used to initiate a quarantine colony. Adult wasps were kept in glass vials (25 mm diameter, 95 mm high) and fed a drop of raw honey spread on the bottom of the foam vial plug. For rearing, fresh (<24 h old) *B. hilaris* eggs were glued to cardboard strips (20 × 60 mm) using Elmer’s washable school glue (Elmer’s Products Inc., Columbus, OH, USA). Egg cards were exposed to mated female wasps in glass vials for 24 h and then left to incubate in the same conditions until adult emergence 24 to 30 days later.

The following experiments used fresh (less than 24 h-old) *B. hilaris* eggs to keep host quality constant, even though *G. aetherium* can successfully parasitize host eggs up to 3–4 days old [18]. All experimental females emerged in rearing vials together with males and/or were housed with an excess of males for at least 24 h before being used in the experiments. All experiments were run in the USDA-ARS quarantine facility in Albany, California, under 22–27 °C, 40–60% RH, and 14 L:10 D photoperiod, except for the assessment of the effect of temperature (see below).

### 2.2. Oviposition Rate and Sex Allocation Sequence

To assess the *G. aetherium* oviposition rate and sex allocation sequence, 22 female wasps, ages varying from a few hours to approximately 1 week old, were released individually in a small petri dish (50 × 9 mm) containing an egg card (20 × 40 mm) with 5 *B. hilaris* eggs. To reflect the ‘single egg’ laying behavior of *B. hilaris*, the eggs were glued in a quincunx pattern to maximize the distance between two eggs (no less than 15 mm) within the small observation arenas. Both naïve and experienced (24 h with 25 eggs) females were tested. The number and sequence of eggs encountered and parasitized were recorded through direct observation using a microscope and a chronometer. The observation stopped as soon as the five eggs on the card were parasitized or at least examined because not all eggs were accepted for oviposition. *Gryon aetherium* females mark their hosts after oviposition by rubbing the tip of their ovipositor against the egg surface [21], which is a reliable indicator of successful oviposition in scelionids [22,23]. The time spent parasitizing eggs on the card, starting at the first egg encountered and ending after the last egg has been marked by the female, was recorded. If no parasitism occurred, the observation stopped after 20 min. After the exposure, all parasitized eggs were incubated separately in microcentrifuge vials until the emergence of parasitoid adults. Unhatched eggs were dissected to record any dead *B. hilaris* nymphs or *G. aetherium* adults. The oviposition rate was calculated as the number of eggs parasitized per h.

### 2.3. Female Density and Duration of Exposure to Hosts

These two factors were investigated together using 1–4-day old *G. aetherium* females and exposing groups of either 1, 2, 5, or 10 females to 20 *B. hilaris* eggs glued to cardboard strips (= egg cards) for 1, 2, or 3 days. Wasps were released in glass vials containing the egg cards with a drop of raw honey spread on the stopper as a food source. At the end of the exposure, the egg cards were incubated at room temperature until wasp emergence stopped. Unhatched eggs were dissected to record any dead *B. hilaris* nymphs or *G. aetherium* adults. The sex ratio was calculated per vial as the number of emerged and dead unemerged females divided by the total number of emerged and dead unemerged males and females. Replicates that did not produce any *G. aetherium* offspring were excluded from the analysis.

### 2.4. Maternal Age/Host Deprivation

The following experiment looked at the combined effect of maternal age and host deprivation: naïve parasitoid females, ages ranging from 1 to 28 days old, were exposed singly to 20 *B. hilaris* eggs for 3 h. This exposure was repeated the following day using 20 fresh *B. hilaris* eggs. The reduction of exposure time from 24 h in the previous experiment to 3 h was to ensure that we provided an excess of hosts (see Table 1 for an assessment of *G. aetherium* oviposition rates), allowing for an estimate of progeny production in addition to progeny sex ratio. Before the exposure, the females were kept in holding vials at a maximum density of five females per vial, with an equal number of males. The exposures took place in small Petri dishes with one small drop of raw honey spread on the floor of the Petri dish as a food source. Female wasps were released on the egg cards in close proximity to the host eggs (1–2 mm). Between exposures, the experimental females were kept singly in a glass vial with 2 males and a drop of raw honey spread on the stopper. After the exposure, the egg cards were incubated at room temperature until wasp emergence. Unhatched eggs were dissected to record any dead *B. hilaris* nymphs or *G. aetherium* adults.

### 2.5. Temperature during Oviposition

The effect of the temperature during oviposition on progeny production and the sex ratio was investigated similarly to the effect of maternal age/host deprivation: 1-day-old naïve females were exposed singly to 20 host eggs for 3 h at different temperatures (19.0, 22.6, 26.6, and 28.9 °C). Egg cards and experimental females were preconditioned at the experimental temperature for 90 min and 30 min, respectively. The 4 temperature treatments were run on 2 consecutive days (2 treatments each day), and there were 11 replicates for each temperature treatment. After the exposure, the egg cards were incubated at room temperature (21.6 ± 0.7 °C, min = 20.2 °C, max = 23.7 °C) until wasp emergence.

### 2.6. Data Analysis

For the density/duration of exposure and maternal age experiments, mixed-model generalized linear models (GLM) were used to analyze the effects on offspring production (using Poisson errors) and proportions of females (using binomial errors and a logit link function) using the glmer function and the lme4 and car packages in R version 4.0.2 [24]. For the density/duration of the exposure experiment, the density and duration of exposure were included as fixed factors, including a density*duration interaction term and trial date as a random factor. For the maternal age experiment, the maternal age and exposure (first/second exposure) were included as fixed factors with a maternal age*exposure interaction term, and with female and trial date as random factors. For the temperature experiment, the temperature was included as a fixed factor in GLMs, using the glm function in R. The GLM model for the sex ratio failed to converge, however, and the logit-transformed proportion of females was compared among temperatures using analysis of variance and the lm function in R. When results for a factor were significant, Tukey HSD tests were used for multiple comparisons using the pairs function in the emmeans package in R.

## 3. Results

### 3.1. Oviposition Rate and Sex Allocation Sequence

Out of the 22 females observed, the 2 youngest females (who emerged the same day) did not parasitize any eggs during the 20 min observation (Table 1). The average oviposition rate of the remaining females was 11.7 ± 4.7 eggs/h (min = 2.6/max = 20.0). Three females produced only male offspring while two females produced only female offspring. Looking at the remaining 15 replicates that produced both male and female offspring, the first male egg was deposited first and second in the oviposition sequence, nine and six times, respectively.

### 3.2. Female Density and Duration of Exposure

On average, progeny production was 14.9 ± 0.5 offspring per replicate with no effect of female density nor duration of exposure (female density: χ^2^ = 4.03, df = 3, *p* = 0.26; duration of exposure: χ^2^ = 1.69, df = 1, *p* = 0.19; female density*duration of exposure: χ^2^ = 0.75, df = 3, *p* = 0.86). There was a significant effect of the duration of exposure on the progeny sex ratio (χ^2^ = 6.13, df = 2, *p* = 0.047) but no significant effect of female density (χ^2^ = 4.68, df = 3, *p* = 0.20) and no interaction between the two factors (χ^2^ = 9.60, df = 6, *p* = 0.14) (Figure 1). The results of a Tukey HSD test showed a significant difference in the progeny sex ratio between 1 d and 2 d of exposure (*p* = 0.02) but not between 1 d and 3 d (*p* = 0.056).

### 3.3. Maternal Age/Host Deprivation

The number of offspring produced first increased with maternal age/host deprivation, from an average of 3.2 offspring (per 3 h of exposure) for 1 d old females to 6.8 offspring for 7 d old females (Figure 2). Not all freshly emerged females (1 d old) produced offspring, 78% and 67% on the first and second exposure, respectively. Progeny production remained relatively constant up to 18 d old females, at about 6.5 offspring, before dropping for 24+ d old females, and was only marginally affected by maternal age (χ^2^ = 17.83, df = 10, *p* = 0.06), but was significantly affected by exposure (χ^2^ = 5.81, df = 1, *p* = 0.02) and the maternal age*exposure interaction (χ^2^ = 37.55, df = 9, *p* < 0.001). When exposures were analyzed separately, offspring production was affected by maternal age in the first exposure (χ^2^ = 32.43, df = 10, *p* < 0.001) but not in the second (χ^2^ = 9.15, df = 9, *p* = 0.42). In the first exposure, the results of a Tukey HSD test showed that offspring production by 28 d old females was lower than 6 (*p* = 0.049), 7 (*p* = 0.04), 11 (*p* = 0.01), and 15 d (*p* = 0.03) old females. Overall, offspring production was lower after the second exposure, except for 2 d old females and 24+ d old females when more progeny was produced on the second day of exposure. The offspring sex ratio was highly variable (Figure 2), and no significant effect of maternal age/host deprivation or exposure time was detected (maternal age: χ^2^ = 3.45, df = 9, *p* = 0.94; exposure time: χ^2^ = 0.49, df = 1, *p* = 0.48; maternal age*exposure: χ^2^ = 6.30, df = 9, *p* = 0.71).

### 3.4. Temperature during Oviposition

There was a significant effect of temperature on offspring production (χ^2^ = 57.10, df = 3, *p* < 0.001), which differed among all temperatures (Tukey HSD test, *p* < 0.05), with the exception of 22.6 and 28.9 °C (Tukey HSD test, *p* > 0.05). Progeny production by *G. aetherium* tended to increase with temperature, from an average of 2.5 at 19.0 °C to 9.4 at 26.6 °C but dropped to 6.4 at 28.9 °C (Figure 3). The offspring sex ratio was not significantly affected by temperature (F_3,21_ = 1.44, *p* = 0.26) (Figure 3).

## 4. Discussion

Our first experiment showed that *G. aetherium* females lay at least one male egg early in their oviposition sequence (first or second egg), similar to other *Gryon* spp. [23,25,26,27]. This strategy can ensure that there will be at least one male offspring irrespective of the host patch size, especially for host species that tend to lay their eggs in patches of varying sizes [26]. This strategy seems well suited for *G. aetherium* since *B. hilaris* is known for its unusual oviposition behavior, where females bury their eggs individually or in small groups in the soil [28], making patch size assessment even trickier for the foraging females.

Progeny production was influenced by maternal age/host deprivation. Assuming the number of eggs laid closely reflects the *G. aetherium* egg load as in *G. pennsylvanicum* [29], the increase in offspring production with female age/host deprivation suggests that naïve females keep maturing eggs for about a week after emergence. The eggs remain viable and ready to be oviposited as soon as hosts become available for about 3 weeks after emergence, although it is possible that this assumed constant egg load is the result of well-balanced rates of egg maturation and egg resorption. The significant decrease in progeny production for 24+ d old females could be a consequence of decreased vitality of older eggs, the onset or increase of egg resorption, and/or the onset of reproductive senescence in geriatric females in general. This effect of host deprivation on progeny production is very similar to another study involving *G. pennsylvanicum* where parasitism rates increased until 10 d of host deprivation, slowly decreasing thereafter [29], and is indicative of a synovigenic mode of reproduction, i.e., when the females keep maturing eggs through her adult life. Other species of egg parasitoids attacking heteropteran host species, including other Scelionidae, also exhibit the synovigenic reproductive strategy [19,30,31,32]. In addition, not all females parasitized hosts soon after emergence (up to 2 d for some, see Table 1 and Figure 2) suggesting that *G. aetherium* females may experience a short pre-oviposition period, also a characteristic of synovigenic parasitoids [33]. Similarly, a short pre-oviposition period of 1.3 ± 0.35 d has been recorded in *G. gallardoi* (Brethes) (Hymenoptera: Scelionidae) [19].

There was no effect of female age on progeny production on the second day of exposure to hosts when progeny production was usually lower, suggesting that, in this context of high host availability and short exposure (3 h twice over a 48 h period), egg maturation may not be fast enough to balance out oviposition, leading to fewer eggs available the second day of oviposition. A study with *G. pennsylvanicum* also showed that naïve females had higher parasitism rates than experienced females and that all females experience transient egg limitation, occurring when egg depletion through oviposition exceeds the rate of egg maturation [34]. Older females may have fewer eggs readily available for oviposition either because of decreased egg viability or resorption, and it may take about 24 h to produce more eggs. Whatever the physiological mechanism used to maintain their egg load, the ability of naïve females to readily parasitize hosts after several weeks of host deprivation is remarkable. A recent study showed that *G. aetherium* females (identified as *G. gonikopalense*, see [35]) may retain this ability even after 2 months of host deprivation [36].

The duration of exposure to hosts affected the offspring sex ratio in our study, with a higher proportion of females emerging after one day of exposure, although the differences were only significant between 1 d and 2 d of exposure (the difference between the 1 and 3 d treatments was marginally not significant). Long access to the same hosts can lead to superparasitism, which in turn can lead to a shift in the offspring sex ratio [2,37,38]. Superparasitism is uncommon in *G. aetherium* females when the time elapsed since the first oviposition is short (1 h) [21]. However, when the time interval is larger (24 h), superparasitism increases from 5% to 33%, possibly because the detectability of the mark left by the ovipositing females decreases with time [21]. It is unclear what mechanism is responsible for the shift in the sex ratio observed in our experiment. Only a look at the primary and secondary sex ratio (i.e., the sex ratio at oviposition and emergence, respectively) could give us some more insight since a difference between the two would indicate a physiological constraint (i.e., differential survival of the sexes in superparasitized hosts) rather than a behavioral decision made by the mother. It is also possible that both mechanisms are in action as observed in *Anaphes victus* Huber (Hymenoptera: Myrmaridae) [14]. In this species, females lay more sons in superparasitized hosts, but that change in the sex ratio is compensated by higher mortality of male larvae in superparasitized hosts, resulting in no significant difference between sex ratio progeny in parasitized and superparasitized hosts. Unfortunately, to our knowledge, there is no way to tell the sex of the egg being oviposited in *G. aetherium*, preventing the comparison of primary and secondary sex ratio.

Female density on the egg card did not significantly influence the progeny sex ratio. In contrast, a shift in the sex ratio, e.g., an increase in male production when female density increases, was observed in other solitary egg parasitoids [39,40,41,42,43]. However, *B. hilaris* differs from other stink bug hosts in that females do not lay eggs in masses on host plants, but instead, lay eggs singly or in small groups in the soil. In nature, there is probably a significant amount of male dispersal before mating occurs, diverging from one of the prerequisites for the local mate competition. Female age/host deprivation also did not affect the offspring sex ratio. This is in accordance with Martel and Sforza [36] who also showed that there were no changes in offspring sex ratio in *G. aetherium* after 30 and 60 d of host deprivation. Similarly, there were no significant changes in the sex ratio in *G. pennsylvanicum* after periods of host deprivation ranging from 5 to 50 d [29].

An effect of temperature on the progeny production and sex ratio of parasitoids is generally observed at extreme temperatures, usually below 15 °C or above 30 °C (see for example [6]). However, testing extreme temperatures was beyond the scope of this study as we just focused on optimizing laboratory conditions. We only investigated a narrow range of milder temperatures (19–29 °C) because lower and higher temperatures would put unnecessary stress on the foraging females, likely leading to lower numbers of eggs parasitized and limiting our sex ratio data set. Additionally, our study differs from most studies assessing the effect of temperature on the reproductive characteristics of parasitoids in that the temperature treatments were applied during oviposition only. Our results showed that the temperature experienced during oviposition affected progeny production by *G. aetherium*, which peaked at 26.6 °C but did not affect the sex ratio. Similarly, the parasitism rate by the egg parasitoid *Anaphes listronoti* Huber (Hymenoptera: Mymaridae) reached a maximum when foraging at 28.4 °C, but the offspring sex ratio did not change with the temperature [44]. Our study is preliminary but suggests that temperatures around 26 °C during oviposition may be optimal for the maximum production of *G. aetherium*. However, further investigations are needed to look at the effect of temperatures on *G. aetherium* development, survival, and fecundity parameters before a final recommendation can be made. Both constant and fluctuating temperature regimes should be considered as recent studies on the bioecology and reproductive biology of mass-produced endo- and ectoparasitoids provide strong evidence that fluctuating temperatures may improve some life-history parameters compared to constant regimes with corresponding means [45,46].

In general, the progeny sex ratio varied considerably in our study, with several females producing only male offspring, suggesting that mating was sometimes unsuccessful or entirely lacking even though the experimental females (1) usually emerged in vials already containing freshly emerged males (protandry) and (2) were housed with an excess of males the day of their emergence and for at least 24 h before being used in the experiment. The males used were young (approx. 1–5 d old), collected from vials from the same rearing dates or rearing vials set up a couple of days later. We avoided releasing males in the experimental vials as they tend to disrupt oviposition and can lower fecundity [18]. Additionally, mating typically occurs once in hymenopteran parasitoids [47,48,49], and our experimental females would have had plenty of opportunities to mate within 24 h after emergence. Most likely, our colony may be experiencing a phenomenon called “pseudovirginity”. In species that mate once in their lifetime, females become unreceptive to males after courtship. In pseudovirginity, females become unreceptive after courtship but before insemination, either because the courtship has been interrupted by another male or because the male has run out of sperm [50,51]. It is unlikely that the males used in our experiment had already run out of sperm, but we cannot rule out the detrimental effect of inbreeding on their fertility. We estimated that our colony may have been through 75 generations since its collection in 2016.

In conclusion, our results indicate that the production of *G. aetherium* can be maximized by limiting the exposure to the same hosts to 24 h and using females that are less than 24 d old, at temperatures around 26 °C.

## Figures and Tables

**Figure 1 insects-13-01010-f001:**
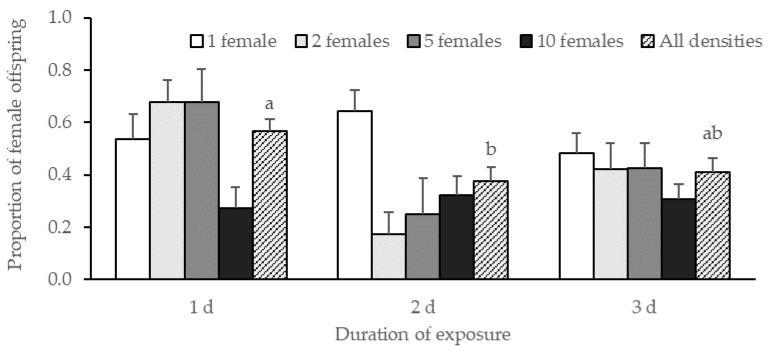
Proportion of *G. aetherium* female offspring ± SE produced at different female densities after 1, 2, or 3 d of exposure to *B. hilaris* eggs. Bars with different letters are significantly different (Tukey’s HSD, *p* < 0.05).

**Figure 2 insects-13-01010-f002:**
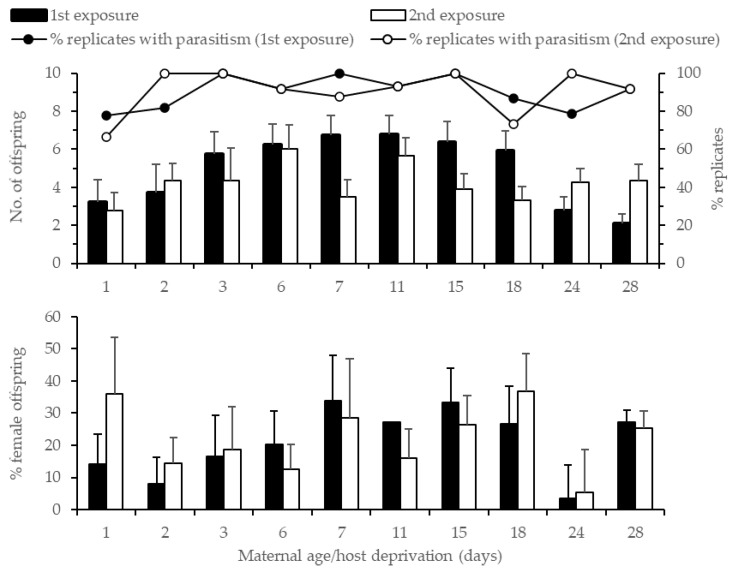
Percent replicates with parasitism, average number of offspring produced (**top**), and offspring sex ratio (**bottom**) ± SE of host-deprived *G. aetherium* females when exposed to 20 *B. hilaris* eggs for 3 h and 2 d consecutively.

**Figure 3 insects-13-01010-f003:**
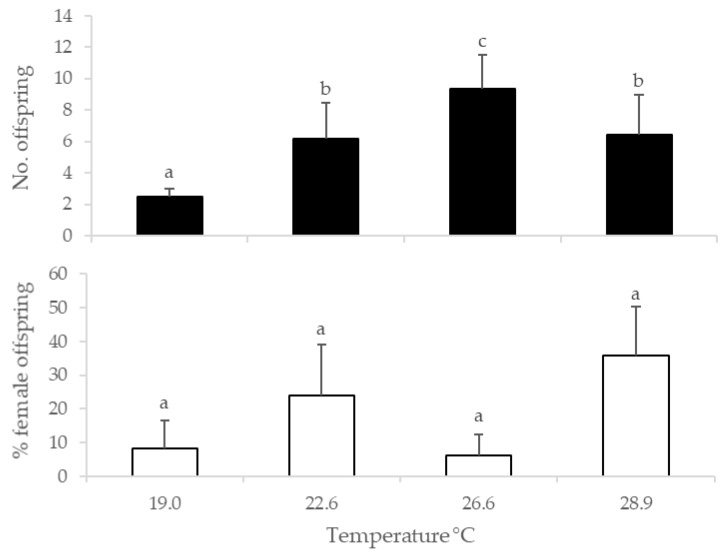
Progeny production (**top**) and sex ratio (**bottom**) of *G. aetherium* exposed at different temperatures during oviposition. Bars with different letters are significantly different (Tukey’s HSD, *p* < 0.05).

**Table 1 insects-13-01010-t001:** Number of eggs parasitized, time spent parasitizing, and position of the first male egg in the oviposition sequence of naïve and experienced *G. aetherium* females of different ages.

Female Age (d)	Experience	No. Eggs Parasitized	Time (min)	No. Eggs/h	Male Offspring Order in Oviposition Sequence
<1	no	0	20	-	-
<1	no	0	20	-	-
1	no	5	32.75	9.2	1
1	no	5	17.75	16.9	1
1	no	5	20.00	15.0	All females
1	no	5	25.25	11.9	1
1	no	5	30.75	9.8	All females
1	no	5	25.50	11.8	1
1	no	5	17.50	17.1	All males
2	yes	5	40.25	7.5	1
2	yes	5	15.00	20.0	2
2	yes	5	35.00	8.6	2
2	yes	5	26.5	11.3	1
2	yes	5	43.25	6.9	1
4	no	5	36.00	8.3	2
4	no	2	47.00	2.6	1
4	no	5	33.00	9.1	All males
7+	yes	5	17.50	17.1	2
7+	yes	5	22.75	13.2	2
7+	yes	3	19.50	9.2	1
7+	yes	3	9.00	20.0	2
7+	yes	3	20.50	8.8	All males
			Average ± SE:	11.7 ± 4.7	

## Data Availability

The data presented in this study are available on request from the corresponding author.

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
