# Peer review of "Factors Affecting Progeny Production and Sex Ratio of Gryon aetherium (Hymenoptera: Scelionidae), a Candidate Biological Control Agent for Bagrada hilaris (Hemiptera: Pentatomidae)"

_insects, 2022, doi:10.3390/insects13111010_

Round 1
Reviewer 1 Report
I recommend acceptance of this manuscript for publication after a few minor edits provided in the attached PDF file. It is a well-written manuscript with important findings regarding mass production of a classical biological control candidate for the bagrada bug. Conditions and factors affecting the reproductive success of this parasitoid species were assessed. The methods applied are appropriate, data are well presented and conclusions are supported by data.

Author Response
I recommend acceptance of this manuscript for publication after a few minor edits provided in the attached PDF file. It is a well-written manuscript with important findings regarding mass production of a classical biological control candidate for the bagrada bug. Conditions and factors affecting the reproductive success of this parasitoid species were assessed. The methods applied are appropriate, data are well presented and conclusions are supported by data.
We thank reviewer 1 for their recommendation. Please find below our point-by-point answer to comments/edits that were provided in a pdf file.
Line 53: we added the two references suggested for the effect of parental age on progeny sex ratio.
Line 160: we removed the redundant “floor”.
Line 303: we replaced “unparasitized” with “superparasitized”
Line 309: we replaced “stinkbugs” with “stink bug”.
Reviewer 2 Report
This article by Evelyne Hougardy and Brian Hogg is well motivated, the structure is appropriate, and the manuscript is well written. The methods used are appropriate for the objectives of the work and, in general, well depicted. The discussion of results and comments on future research should be improved before the paper can be accepted for publication in Insects.
My major concern is that the authors are extrapolating the applicability of their results beyond what the design supports. These are only data from a narrow set of highly artificial laboratory conditions, so the inference power of the paper is very limited, but authors do not acknowledge this detail at all and need to be more forthcoming. More specifically, only a narrow range of constant temperatures was investigated in this study (i.e., 19-29) for progeny production and sex ratios of BCAs. The effects of temperature profiles lower than 19C and higher than 29C on the same parameters of BCAs were not investigated in this study. This is a critical limitation of the study, and the authors must concede and discuss this.
Also, the possibility of using fluctuating temperature profiles was not investigated in this study either. The interaction of cyclic temperatures with nonlinear characteristics of reproductive parameters of BCAs can introduce significant deviations from the results obtained here, especially at the lower and higher temperatures of the reproductive activity functions. Therefore, studies across a broader set of fluctuating temperature regimes are still necessary to understand the real effect of temperature on reproductive characteristics of BCAs, as this is the closest to the daily temperature fluctuations that occur in the field. So, I am suggesting to the authors to tone-down the language a little and admit that there are still substantive uncertainties to be considered.
Some of the authors statements would be much stronger if they tie their work to the body of literature that has built up on the bioecology and reproductive biology of other mass-produced endo- and ectoparasite biocontrol agents (BCAs) for field releases in California. They all point to the same direction and should be paired back to this study. Some examples are J. Econ. Entomol. 112: 1560-1574 (mass produced ectoparasite BCAs) or J. Econ. Entomol. 112:1062-1072 (mass produced endoparasite BCAs), but there are others too. These studies provide strong evidence of increased longevity in BCAs reared at non-stressful low temperatures when compared to higher temperature regimes. Adding these details will improve the paper in my opinion. They further suggest that the parasitism or egg load was significantly higher at intermediate temperatures (20-30C) than at cline margins (<15C or >35C). This article should provide details on all these fronts to provide the proper context for the work. This is not to diminish the data gathered in this study, they are of value. But it is important for the authors not to overgeneralize, and to warn the reader, including regulatory agencies, against doing so as well.
Good luck!
Author Response
This article by Evelyne Hougardy and Brian Hogg is well motivated, the structure is appropriate, and the manuscript is well written. The methods used are appropriate for the objectives of the work and, in general, well depicted. The discussion of results and comments on future research should be improved before the paper can be accepted for publication in Insects.
My major concern is that the authors are extrapolating the applicability of their results beyond what the design supports. These are only data from a narrow set of highly artificial laboratory conditions, so the inference power of the paper is very limited, but authors do not acknowledge this detail at all and need to be more forthcoming. More specifically, only a narrow range of constant temperatures was investigated in this study (i.e., 19-29) for progeny production and sex ratios of BCAs. The effects of temperature profiles lower than 19C and higher than 29C on the same parameters of BCAs were not investigated in this study. This is a critical limitation of the study, and the authors must concede and discuss this.
Yes, an effect of temperature on progeny production and sex ratio is generally observed at extreme temperatures, usually below 15 °C or above 30 °C. However, testing extreme temperatures was beyond the scope of this study as we just focused on optimizing laboratory conditions. We investigated a narrow range of temperature because extreme temperatures would put an unnecessary stress on the foraging females and likely affect parasitism level, with the risk of not having enough data for a sex ratio estimate. In response to this comment, we’ve added a few sentences in the discussion to remind the reader of the limitations of our study:
“An effect of temperature on the progeny production and sex ratio of parasitoids is generally observed at extreme temperatures, usually below 15 °C or above 30 °C (see for example [6]). However, testing extreme temperatures was beyond the scope of this study as we just focused on optimizing laboratory conditions. We only investigated a narrow range of milder temperature (19 – 29 °C) because lower and higher temperatures would put an unnecessary stress on the foraging females, likely leading to lower numbers of eggs parasitized and limiting our sex ratio data set.”
Also, the possibility of using fluctuating temperature profiles was not investigated in this study either. The interaction of cyclic temperatures with nonlinear characteristics of reproductive parameters of BCAs can introduce significant deviations from the results obtained here, especially at the lower and higher temperatures of the reproductive activity functions. Therefore, studies across a broader set of fluctuating temperature regimes are still necessary to understand the real effect of temperature on reproductive characteristics of BCAs, as this is the closest to the daily temperature fluctuations that occur in the field. So, I am suggesting to the authors to tone-down the language a little and admit that there are still substantive uncertainties to be considered.
Our temperature treatments were only applied for a short time (3 h exposure to host eggs) so testing fluctuating temperature regimes didn’t make much sense in this context. We understand the confusion as our study is a little different from most temperature studies out there. We’ve added a sentence in the discussion to remind the reader of the particulars of our study:
“Also, our study differs from most studies assessing the effect of temperature on re-productive characteristics of parasitoids in that the temperature treatments were applied during oviposition only.”
In addition, we are now providing more details in the Materials and Methods about the room temperature experienced during incubation of the parasitized eggs. Because the four temperature treatments were run almost simultaneously (on two consecutive days), the parasitized eggs experienced the same temperature conditions during their incubation. We are now providing specifics on the room temperature in the quarantine at the time of the study. We’ve added the following sentences (the new information is underlined here):
“The four temperature treatments were run on two consecutive days (2 treatments each day) and there were 11 replicates for each temperature treatment. After the exposure, the egg cards were incubated at room temperature (21.6 ± 0.7 °C, min = 20.2 °C, max = 23.7 °C) until wasp emergence”.
Now, we agree with the reviewer that our study was preliminary and that much more needs to be done to assess the effect of temperature on other reproductive characteristics of BCAs (development time, longevity, fecundity), including using fluctuating temperature regimes. We’ve added a sentence in the discussion to address this comment:
“Further investigations are needed to look at the effect of temperature on G. aetherium development, survival and fecundity parameters before a final recommendation can be made.”
Some of the authors statements would be much stronger if they tie their work to the body of literature that has built up on the bioecology and reproductive biology of other mass-produced endo- and ectoparasite biocontrol agents (BCAs) for field releases in California. They all point to the same direction and should be paired back to this study. Some examples are J. Econ. Entomol. 112: 1560-1574 (mass produced ectoparasite BCAs) or J. Econ. Entomol. 112:1062-1072 (mass produced endoparasite BCAs), but there are others too. These studies provide strong evidence of increased longevity in BCAs reared at non-stressful low temperatures when compared to higher temperature regimes. Adding these details will improve the paper in my opinion. They further suggest that the parasitism or egg load was significantly higher at intermediate temperatures (20-30C) than at cline margins (<15C or >35C). This article should provide details on all these fronts to provide the proper context for the work. This is not to diminish the data gathered in this study, they are of value. But it is important for the authors not to overgeneralize, and to warn the reader, including regulatory agencies, against doing so as well.
We added a few sentences in the discussion to tie our research with the recent studies mentioned by the reviewer:
“Our study is preliminary but suggests that temperatures around 26 °C during oviposition may be optimal for maximum production of G. aetherium. However, further investigations are needed to look at the effect of temperature on G. aetherium development, survival and fecundity parameters before a final recommendation can be made. Both constant and fluctuating temperature regimes should be considered as recent studies on the bioecology and reproductive biology of mass-produced endo- and ectoparasitoids provide strong evidence that fluctuating temperatures may improve some life-history parameters compared to constant regimes with corresponding means [45,46].”
Reviewer 3 Report
The authors conducted a series of simple experiments aimed to estimate the effects of temperature, female density, duration of exposure, maternal age and host deprivation on progeny production by an insect parasitoid Gryon aetherium, a potential agent for biological control of a bug pest Bagrada hilaris. The experiments were well designed and the data were mostly correctly analyzed. The results of the study can be used for the elaboration of the methods for mass and laboratory rearing of the parasitoid. Thus, the manuscript can be published, although a substantial revision (including not only minor corrections but also re-analysis of some data) is necessary before publication (see my comments below).
Line 143: please, indicate the proportion of honey and water in the solution.
Lines 146-148: it is stated that “the sex ratio was calculated per vial as the number of emerged and dead unemerged females divided by the total number of emerged and dead unemerged males and females” whereas in line 177 the proportion of (only?) emerged females is mentioned.
First, please, resolve this contradiction. Moreover, it should be clearly indicated if unemerged (dead) progeny was also included in the total number of offspring produced.
Second, the data on the proportion of dead unemerged progeny adults should be also analyzed: please, give the average values and estimate the significance of the effect of experimental factors on this parameter (as you did for the total number of progeny, percent females etc.).
Lines 196-197: please, estimate the significance of these data: did females really tended to start with a male egg more often than with a female egg? You can, for example, compare distributions of males and females over the first 5-10 eggs.
Line 208: please, indicate the exact value of the difference between 1 d and 2 d of exposure
Lines 210-211: I guess that different letters above the bars (Fig. 1) indicate significant difference between the data for different exposures, but this should be indicated in the legend.
Line 224: again (same as in line 208) give the exact value of significance of the difference by the Tukey HSD test (as you do for the shi-square test).
Figure 2: for the proportion of female offspring, SE are rather often close to or even higher than corresponding means suggesting that distribution of these data were very far from being normal. In such cases, use of other descriptive statistics, such as medians and quartiles is recommended.
Lines 240-241: as seen in Fig. 3, the difference in the proportion of females between temperatures was not statistically significant (all bars are labeled with the same letter) and therefore it is not reasonable (and even not correct) to indicate at what temperatures the highest and the lowest values were observed.
Lines 242-243: please, explain in the legend the meaning of the letters above the bars (see my comment to lines 210-211).
Figures 2 and 3: it seems that the same parameters are indicated along the vertical axes with slightly different terms. “No. of offspring” is the same as “No. of offspring produced” and “% female offspring” is the same as “percent females”. This is confusing. Please, keep the same terminology over the whole paper and figures.
Lines 245-246: This conclusion is not valid as the significance of this result was not estimated (see my comment to lines 196-197).
Line 289: As far as I understand, the difference between 1 d and 3 d of exposure was marginally NOT significant (see line 208).
Line 323: I guess, you mean either “temperatures from 26 to 28” or “temperatures of 26–28” (as in line 345).
Author Response
The authors conducted a series of simple experiments aimed to estimate the effects of temperature, female density, duration of exposure, maternal age and host deprivation on progeny production by an insect parasitoid Gryon aetherium, a potential agent for biological control of a bug pest Bagrada hilaris. The experiments were well designed and the data were mostly correctly analyzed. The results of the study can be used for the elaboration of the methods for mass and laboratory rearing of the parasitoid. Thus, the manuscript can be published, although a substantial revision (including not only minor corrections but also re-analysis of some data) is necessary before publication (see my comments below).
Line 143: please, indicate the proportion of honey and water in the solution.
We used undiluted raw honey for rearing and throughout our experiments. We corrected the text whenever a honey/water solution was mentioned.
Lines 146-148: it is stated that “the sex ratio was calculated per vial as the number of emerged and dead unemerged females divided by the total number of emerged and dead unemerged males and females” whereas in line 177 the proportion of (only?) emerged females is mentioned.
First, please, resolve this contradiction. Moreover, it should be clearly indicated if unemerged (dead) progeny was also included in the total number of offspring produced.
Yes, both progeny production and percent female offspring included dead unmerged wasps (see response to next comment to see what we are referring to). We corrected line 177 and removed the word “emerged”.
Second, the data on the proportion of dead unemerged progeny adults should be also analyzed: please, give the average values and estimate the significance of the effect of experimental factors on this parameter (as you did for the total number of progeny, percent females etc.).
By “dead unemerged”, we are referring to fully developed adult wasps that failed to emerge from the host egg. The most likely explanation for this adult mortality is the presence of excess glue residue (from the glue we used to secure the eggs to the egg card) on the outer surface of the host eggs. The eggs are usually covered in sand grains and it is possible that the added glue created an additional layer the emerging parasitoid has sometimes difficulty chewing through. Thus, inability of adults to emerge is probably an artefact of using glue, and is unlikely to have biological relevance. Unfortunately, we did not keep track of this mortality consistently for all the experiments but estimated that about 9% of the parasitoids died while trying to emerge from the host eggs (data from the temperature experiment only). We made a note to look at this phenomenon more closely during future investigations.
Lines 196-197: please, estimate the significance of these data: did females really tended to start with a male egg more often than with a female egg? You can, for example, compare distributions of males and females over the first 5-10 eggs.
This is an interesting suggestion, but we do not believe our data would allow us to examine whether male eggs were more likely to be laid first. The purpose of these observations was merely to determine whether female G. aetherium laid male eggs early in the oviposition sequence (i.e., as the first or second egg).
Line 208: please, indicate the exact value of the difference between 1 d and 2 d of exposure
We provided the exact P-value from the Tukey HSD test in the text.
Lines 210-211: I guess that different letters above the bars (Fig. 1) indicate significant difference between the data for different exposures, but this should be indicated in the legend.
We added in the figure caption: “Bars with different letters are significantly different (Tukey’s HSD, P < 0.05).”
Line 224: again (same as in line 208) give the exact value of significance of the difference by the Tukey HSD test (as you do for the shi-square test).
We provided the exact P-values from the Tukey HSD test.
Figure 2: for the proportion of female offspring, SE are rather often close to or even higher than corresponding means suggesting that distribution of these data were very far from being normal. In such cases, use of other descriptive statistics, such as medians and quartiles is recommended.
We agree that proportions of female offspring were highly variable, as we mention in the Results section. However, for ease of interpretation we prefer to follow the typical format for presenting proportions, and present them as means ± SE.
Lines 240-241: as seen in Fig. 3, the difference in the proportion of females between temperatures was not statistically significant (all bars are labeled with the same letter) and therefore it is not reasonable (and even not correct) to indicate at what temperatures the highest and the lowest values were observed.
We removed the second part of the sentence comparing the results and a similar statement in the discussion. We also changed our recommended temperature to “temperatures around 26 °C” instead of “temperatures from 26 to 28 °C”.
Lines 242-243: please, explain in the legend the meaning of the letters above the bars (see my comment to lines 210-211).
We added in the figure caption: “Bars with different letters are significantly different (Tukey’s HSD, P < 0.05).”
Figures 2 and 3: it seems that the same parameters are indicated along the vertical axes with slightly different terms. “No. of offspring” is the same as “No. of offspring produced” and “% female offspring” is the same as “percent females”. This is confusing. Please, keep the same terminology over the whole paper and figures.
We changed the y-axis titles in Figure 3 to match the terminology used in Figure 2.
Lines 245-246: This conclusion is not valid as the significance of this result was not estimated (see my comment to lines 196-197).
As mentioned above, the purpose of our observations on oviposition sequence was to determine whether male eggs were laid early in the oviposition sequence, not to determine whether male eggs were more likely to be laid first. To make this clearer, we changed the first sentence of the Discussion to:
“Our first experiment showed that G. aetherium females lays at least one male egg early in their oviposition sequence (first or second egg), similarly to other Gryon spp.”
Line 289: As far as I understand, the difference between 1 d and 3 d of exposure was marginally NOT significant (see line 208).
Correct. We added “not” in the text.
Line 323: I guess, you mean either “temperatures from 26 to 28” or “temperatures of 26–28” (as in line 345).
We changed this phrase to “at temperatures around 26 °C” everywhere in the text in response to the comment for line 240-241. Since our results showed that the sex ratio is not significantly affected by temperatures, we cannot claim that a temperature of 28 °C is preferable despite what Figure 3 is showing. Our results only support that, among the temperatures tested, a temperature around 26 °C is optimum for progeny production.
Round 2
Reviewer 2 Report
Authors have done a great job addressing all of my original comments and those of other reviewers. I have no further suggestions to improve the paper. Thank you.
Reviewer 3 Report
The manuscript was substantially improved. In particular, the authors considered all my comments. Thus, the paper can be published.